# Comparison and Identification for Rhizomes and Leaves of *Paris yunnanensis* Based on Fourier Transform Mid-Infrared Spectroscopy Combined with Chemometrics

**DOI:** 10.3390/molecules23123343

**Published:** 2018-12-17

**Authors:** Yi-Fei Pei, Qing-Zhi Zhang, Zhi-Tian Zuo, Yuan-Zhong Wang

**Affiliations:** 1Institute of Medicinal Plants, Yunnan Academy of Agricultural Sciences, Kunming 650200, China; feifei950222@gmail.com; 2College of Traditional Chinese Medicine, Yunnan University of Traditional Chinese Medicine, Kunming 650500, China; ynkzqz@126.com

**Keywords:** *Paris polyphylla* Smith var. *yunnanensis*, multivariate analysis, chemometrics, Fourier transform infrared

## Abstract

*Paris polyphylla*, as a traditional herb with long history, has been widely used to treat diseases in multiple nationalities of China. Nevertheless, the quality of *P. yunnanensis* fluctuates among from different geographical origins, so that a fast and accurate classification method was necessary for establishment. In our study, the geographical origin identification of 462 *P. yunnanensis* rhizome and leaf samples from Kunming, Yuxi, Chuxiong, Dali, Lijiang, and Honghe were analyzed by Fourier transform mid infrared (FT-MIR) spectra, combined with partial least squares discriminant analysis (PLS-DA), random forest (RF), and hierarchical cluster analysis (HCA) methods. The obvious cluster tendency of rhizomes and leaves FT-MIR spectra was displayed by principal component analysis (PCA). The distribution of the variable importance for the projection (VIP) was more uniform than the important variables obtained by RF, while PLS-DA models obtained higher classification abilities. Hence, a PLS-DA model was more suitably used to classify the different geographical origins of *P. yunnanensis* than the RF model. Additionally, the clustering results of different geographical origins obtained by HCA dendrograms also proved the chemical information difference between rhizomes and leaves. The identification performances of PLS-DA and the RF models of leaves FT-MIR matrixes were better than those of rhizomes datasets. In addition, the model classification abilities of combination datasets were higher than the individual matrixes of rhizomes and leaves spectra. Our study provides a reference to the rational utilization of resources, as well as a fast and accurate identification research for *P. yunnanensis* samples.

## 1. Introduction

The perennial herb plant *Paris* is a genus in the Liliaceae family. *Paris* is one of more than 2000 medicinal plants described in the Chinese Pharmacopoeia (2015 edition), and it has utmost important medicinal effects on treating diseases, including snake bite and insect sting, innominate toxin swelling, and various inflammatory and traumatic injuries with ancient history in China. In addition, *Paris* is also used as an ethnobotanical medicinal herb in Nepal and India, which export *Paris* raw materials every year to China to meet the Chinese traditional medicine (TCM) market demand [1]. *Paris* medicinal plants sold in today’s TCM markets were both of wild and cultivated types, with the number of wild *Paris* gradually decreasing, with a long-term growth cycle, immoderate harvesting, and huge commercial activities [2]. Additionally, amongst almost 28 species and varieties of *Paris*, only *Paris polyphylla* Smith var. *chinensis* (Franch.) Hara (*P. chinensis*) and *P. polyphylla* var. *yunnanensis* (Franch.) Hand. -Mazz (*P. yunnanensis*) are officially described by the Chinese Pharmacopoeia (2015 edition), which further restricted the number of *Paris* medicinal plants [3,4,5]. Hence, substitutes with similar medicinal effects and chemical compounds are considered for selection from the closely related species of *P. yunnanensis* and *P. chinensis*, and other parts of the plants, such as stems and leaves. 

A serious problem is that many number of leaves of *Paris* medicinal plants were abandoned every year, with the rhizomes being unable to meet the market demand. Thus, the use of *P. yunnanensis* and *P. chinensis* leaves as substitutes for the primary choice was to be considered. Currently, Qin et al. have reviewed the feasibility for whether renewable above-ground parts (leaves and stems) of *P. yunnanensis* could be used as an alternative source to rhizomes [6]. They concluded that the above-ground parts can be the substitute source for the rhizomes of *P. yunnanensis*, in that similar pharmacological properties, including antimicrobial, hemostatic, cytotoxic, and other effects. A variety of quality of Paridis Rhizomes in TCM markets may affect the quality of Chinese patent medicines based on *P. yunnanensis* rhizomes. On these basis, it is necessary and meaningful to quickly assess the quality of *P. yunnanensis* rhizomes and leaves. 

Yunnan possesses complex climatic conditions, which means that the quality of TCM plants varies with different climatic conditions of different geographical origins in Yunnan. A variety of analytic techniques have been applied to determine the active chemical components and fingerprints to assess the quality of *P. yunnanensis* samples, including ultra-high performance liquid chromatography-mass spectrometry (UHPLC-MS) [7,8], ultraviolet-visible (UV-Vis) [9], high performance liquid chromatography (HPLC) [10], and Fourier transform mid infrared (FT-MIR) [8,11,12], and so on. Up to now, chemometrics has been widely applied to herbal medicines and plant spectral analyses [13,14]. For example, principal component analysis (PCA) often was used to research Chinese herbal medicines of multiple tissues and geographical origins [15,16]. Partial least squares discriminant analysis (PLS-DA) and random forest (RF) have been gradually applied to the field of traditional Chinese herbs in recent years, such as *Panax notoginseng*, *Dendrubium officinale*, etc. [17,18]. Our previous studies have demonstrated that all of these techniques have obtained better identification abilities for *P. yunnanensis* from different geographical origins. Compared with chromatography, the better classification abilities, more convenience, and time-saving techniques were displayed using spectroscopy techniques. To date, combined with various analytical techniques, chemometrics methods have been successfully applied to assess *P. yunnanensis* samples with better classification and identification abilities, including support vector machine [19], RF [11,12], hierarchical cluster analysis (HCA) [10,12,20,21], PLS-DA [9,12,22], and PCA [9,12,22]. However, they failed to analyze other parts of *P. yunnanensis* to fast assess their quality, as well as comparing and combining rhizomes and leaves to identify *P. yunnanensis* from a variety of geographical origins. Hence, the purpose of our study is to assess the quality of *P. yunnanensis* medicinal materials by determining their rhizomes and leaves in FT-MIR spectra, combined with chemometrics.

In this study, to further obtain better, faster, and reliable identification methods for *P. yunnanensis* raw materials from different geographical origins, we investigated *P. yunnanensis* samples from six regions from Yunnan Province by FT-MIR spectroscopy, combined with four chemometrics methods, including PCA, PLS-DA, RF, and HCA. The influence on the fast-quality assessment effects of different parts, including leaves and rhizomes of *P. yunnanensis* were compared. The results may demonstrate the importance of the leaves of *P. yunnanensis*, and they can provide direction for the future development of *P. yunnanensis* medicinal plants.

## 2. Results and Discussion

### 2.1. Comparison Analysis between Rhizomes and Leaves

The raw and SD FT-MIR spectra of rhizomes and leaves of *P. yunnanensis* samples from six geographical regions are showed in Figure 1. The peaks height, character, and position among different geographical origins samples are similarly shown in Figure 1a. Characteristic peaks appeared at ~3328 cm^−1^, and were assigned to O–H absorption, at ~2726, 1414, and 1370 cm^−1^ to methylene and methyl stretching, and bending vibration. Absorption at ~1742 cm^−1^ was endorsed to C=O stretching vibration, at ~1650 cm^−1^ it was attributed to C=C and C=O stretching vibration, which may be attributed to oils, saccharides, steroid saponins, and flavonoids. Besides, absorption at ~1244 cm^−1^ was assigned to C–O stretching vibration, while ~1151, 1078, and 1020 cm^−1^ were endorsed to C–C, C–O stretching vibration and C–OH bending vibration, as well as the main attribute to saccharides and glycosides. Absorption at ~929 cm^−1^ was assigned to the sugar skeleton. These attributes for characteristic peaks were in accordance with studies by Sun et al. and Yang et al. [23,24]. Absorption at ~2855, 1547, 1340, 862, 765, 708, 611, and 580 cm^−1^ were also showed in these FT-MIR spectra. Absorption at ~1650 cm^−1^ and ~1020 cm^−1^ were the key peaks among all absorption peaks of the raw FT-MIR spectra of rhizomes. Additionally, many details of spectral information were shown by standard normal variate–second derivative (SNV-SD) FT-MIR rhizomes spectra in Figure 1c. In detail, among the peaks regions of 1200–900 cm^−1^, the peaks absorptions were at 1173, 1135, 1093, 1065, 1050, 1035, 996, 976, and 950 cm^−1^, which are not shown in the raw FT-MIR spectra of rhizomes.

The raw FT-MIR spectra of leaves showed different peak heights, characters and positions and numbers of the characteristic peaks for those of rhizomes, which are shown in Figure 1b. Compared with the raw rhizomes FT-MIR spectra, absorption for the raw leaves spectra exhibited a red-shift at 1750–1290 cm^−1^, and a blue-shift at 1290–950 cm^−1^. In other words, various differences of chemical information was reflected by the raw rhizomes and leaf FT-MIR spectra. Similar to the raw rhizome FT-MIR spectra, the absorption was mainly attributed to oils, saccharides, steroid saponins, flavonoids saccharides, and glycosides. Namely, absorption at 1602 cm^−1^ and 1053 cm^−1^ are the two key peaks of the raw leaf FT-MIR spectra. Similarly, certain details from the spectral information are shown in SNV-SD leaf FT-MIR spectra in Figure 1d. In detail, among peaks regions of 1200–900 cm^−1^, the peak absorptions at 1187, 1124, 1088, 974, and 938 cm^−1^ are proven, which are not shown in the FT-MIR spectra of raw leaves.

The PCA score plot and loading plot based on the total FT-MIR spectra are shown in Figure 2. Besides, 72.9% and 17% FT-MIR spectra information were exhibited by PC 1 and PC 2, respectively. Two parts (rhizomes and leaves) were well separated by the first two principal components (PCs) in the PCA score plot. Absorption at 1300–550 cm^−1^ by PC 1 contributed to a higher importance than that of PC 2. In other words, the bands of this region are more important to PC 2.

### 2.2. Origin Traceability Based on Chemometrics

#### 2.2.1. Using Rhizome FT-MIR Spectra Datasets

Raw FT-MIR rhizomes spectra were pretreated by SNV, standard normal variate-first-derivative (SNV-FD), SNV-SD, and SD preprocessing methods, and to select the best pretreatment method. All parameters for these pretreatment methods are shown in Appendix A. Comparing parameters to the raw FT-MIR spectra, all parameters are better after preprocessing. Among them, SNV-SD was defined as the optimal preprocessing method for the fundamental for the larger values of cumulative interpretation ability (R^2^), cumulative prediction ability (Q^2^), and accuracy of the calibration set, as well as the lower values of the root mean square error of estimation (RMSEE) and the root mean square error of cross-validation (RMSECV). Despite SD obtaining a better accuracy, SNV-SD obtained a lower RMSEE, RMSECV, and latent variables (LVs). In our following study for rhizomes, models established by raw and the best preprocessing (SNV-SD) FT-MIR spectra data will be compared.

The variable importance for the projection (VIP) scores for values greater than 1 of the raw rhizome FT-MIR data are shown in Figure 3a. The regions of 1750–1500 cm^−1^ and 1200–750 cm^−1^ are important variables regions for differentiating six geographical origins of *P. yunnanensis* by FT-MIR spectra. The bonds at 1750–1500 cm^−1^ are mainly attributed to oils, saccharides, steroid saponins, and flavonoids compounds. Besides, the bands at 1200–750 cm^−1^ are mainly endorsed to saccharides and glycosides compounds. The two key peaks of raw rhizome FT-MIR spectra were contained in these two bands. What’s more, there were also some peaks that were not clearly identified, and these peaks are equally important for the identification of *P. yunnanensis* samples from different origins. On the basis of the SNV-SD rhizome FT-MIR data, the VIP scores for values greater than 1 are shown in Figure 3b. The degrees of important variables regions from 1750–750 cm^−1^ seem to be similar in importance for the differentiation of six geographical origins of *P. yunnanensis* by FT-MIR spectra. It was further demonstrated that each peak was important for distinguishing *P. yunnanensis* samples from different geographical origins.

RF models were established on raw and SNV-SD rhizome FT-MIR spectra data matrixes. The 1207 and 1202 variables were contained in raw and SNV-SD rhizome FT-MIR spectra datasets, respectively. For the two RF models of raw and SNV-SD rhizomes FT-MIR spectra, the initial number of trees (n_tree_) were set as 2000 trees. The suitable value of n_tree_ was selected, based on the lowest total value, and the need to be assured of the lower values of the most classes. The 1328–1392 trees and 650–740 trees are the lowest ranges for n_tree_ of raw and SNV-SD rhizomes datasets, respectively, which are shown in Figure 4a,b. Besides, the optimal values 1383 and 951 trees were obtained for further selection of the suitable number of variable (m_try_) values of the RF models, based on raw and SNV-SD rhizomes FT-MIR datasets, respectively. As shown in Figure 4c,d, the optimal m_try_ were calculated to be 33 and 36, according to the lowest out-of-bag (OOB) values for the raw and SNV-SD datasets, respectively. The suitable n_tree_, combined with the optimal m_try_, were used to select the most important variables.

To start with, all variables of the raw and SNV-SD datasets were sorted from the least important variables, to the most important variables, respectively. The 10-fold cross validation error rates of the RF model, based on raw and SNV-SD FT-MIR datasets of rhizomes *P. yunnanensis* samples are shown in Figure 5a,b. It was reduced sequentially by five variables for each step for the initial variables of 1207 and 1202, for raw and SNV-SD datasets, respectively. In both the range of 1–1207 and 1–1202 variables numbers, all important variables were divided into three regions. Among these regions, the 10-fold cross validation error rate values showed a reduced or incremental trend. When the 10-fold cross validation error rate shows a drop trend and then an upward trend, that number of variables at the turning point is likely to be the optimal number for the most importance variables. Hence, variable numbers of 207 and 292 with a lower than 10-fold cross validation error rate for 0.34202 and 0.08143 were selected, to establish the RF models of raw and SNV-SD rhizome FT-MIR spectra, respectively.

When the most important variables were re-selected, forming the new data matrix, it was necessary for the reconstruction of optimal n_tree_ and m_try_ values for raw and SNV-SD FT-MIR spectra. The selecting process was the same as above. As shown in Figure 6, the 1011–1201 trees and 788–880 trees are the lowest ranges for n_tree_ of raw and SNV-SD rhizomes dataset, respectively. Finally, 1110 and 820 trees are selected for the optimal n_tree_, as well as 19 and 26, are selected for the best m_try_ of raw and SNV-SD FT-MIR rhizome spectra of *P. yunnanensis* samples, respectively. These optimal n_tree_ and m_try_ were used to establish the RF model, and they obtained the accuracy of the calibration set and the validation set, respectively. It is undeniable that the variable selection process is important. The error rate for calibration set of raw datasets was reducing from 36.16% to 33.88%, and it was decreasing from 10.42% to 8.79% for the SNV-SD dataset. In addition, the geographical origin classification ability of the RF model, based on SNV-SD FT-MIR spectra of rhizome *P. yunnanensis* samples, was significantly better than that of the raw spectra.

The parameters for each class of calibration set and validation set of the PLS-DA and RF models, based on raw and SNV-SD rhizomes FT-MIR spectra data matrixes, are shown in Appendix A. The values for all parameters of each class of calibration set and the validation set for the PLS-DA model, based on raw FT-MIR data matrixes, were higher than that of the RF model, and they differ greatly. Additionally, all parameters for the RF model based on SNV-SD FT-MIR data matrixes were greatly enhanced and close to that of the PLS-DA model. Obviously, the parameters of two models for the SNV-SD data matrixes based on FT-MIR spectra were higher than those of raw data matrixes. However, the identification abilities and accuracy for two models based on rhizome FT-MIR spectra were needed for improvement.

#### 2.2.2. Using Leaf FT-MIR Spectra Datasets

Raw leaf FT-MIR spectra dataset was preprocessed by SNV, SNV-FD, SNV-SD, and SD preprocessing methods to select the best pretreatment method. All parameters for these four kinds of preprocessing methods are displayed in Appendix A. Similarity to rhizomes, all parameters for the preprocessed model of FT-MIR spectra for leaves are better than those of the raw data matrix. Besides, the SNV-SD pretreatment among all preprocessing methods was the best one for classifying the different origins of *P. yunnanensis* leaf samples, which possessed values of R^2^, Q^2^, RMSEE, RMSECV, accuracy and LVs that were more satisfactory than other pretreatment methods. For the following study of leaves, models established by raw data, and the best pretreatment (SNV-SD) FT-MIR spectra data were selected to study.

The VIP scores for values greater than 1 of the raw leaf FT-MIR data are shown in Appendix A. The region of 1800–1700 cm^−1^ is the most important variable region for differentiating six geographical origins of *P. yunnanensis* by leaf FT-MIR spectra. The regions of 1700–1300 cm^−1^, 1250–1100 cm^−1^, and 1200–750 cm^−1^ almost possessed equally important degrees for differentiating various geographical origins of *P. yunnanensis* by leaf FT-MIR spectra. The bonds at these regions are also mainly assigned to oils, saccharides, steroid saponins, and flavonoids, saccharides, and glycoside compounds. What’s more, the number of important variables of leaf VIP scores were more than those of rhizome VIP scores, which reflected the difference in chemical information in classifying *P. yunnanensis* samples from different regions. Based on the SNV-SD leaf FT-MIR data, the VIP scores for values greater than 1 are displayed in Appendix A. Compared with the other three regions, variables important for the region of 1800–1700 cm^−1^ show greater importance. Similar, it was also demonstrated that each peak of leaf FT-MIR spectra was important to distinguish *P. yunnanensis* samples from a variety of geographical origins. However, a number of peaks were non-identified chemical compounds in the leaf FT-MIR spectra.

RF models were established on raw and SNV-SD leaf FT-MIR spectra datasets. To start with, the 1207 and 1201 variables were contained in the raw and SNV-SD leaf FT-MIR spectra matrixes, respectively. Similar to the rhizomes, the initial n_tree_ were set as 2000 trees for the RF models of raw and SNV-SD leaf FT-MIR spectra. As shown in Appendix A, 947–961 trees and 980–1008 trees were selected to be the lowest ranges for n_tree_ of raw and SNV-SD leaf datasets, respectively. Additionally, the optimal values of 951 and 982 trees were selected for further selection of the suitable m_try_ values of RF models, based on raw and SNV-SD leaf FT-MIR datasets, respectively. As shown in Appendix A, the optimal m_try_ were calculated to be 42 and 31, respectively.

Like rhizomes, all variables of the raw and SNV-SD matrixes of leaves were ranked from to the least important variables to the most important variables, respectively. The 10-fold cross-validation error rates of the RF model, based on the raw and SNV-SD FT-MIR datasets of leaf *P. yunnanensis* samples are shown in Appendix A. In addition, in both the range of 1–1207 and 1–1201 variables numbers, all important variables, were also divided into three regions. Moreover, variable numbers of 157 and 441 with lower than 10-fold cross validation error rates for 0.36808 and 0.02280 were selected to establish the RF models of the raw and SNV-SD FT-MIR spectra, respectively. The 10-fold cross validation error rate of the SNV-SD matrix was far below that of the raw dataset.

Similar to rhizomes, the most important variables were as the new data matrixes, and meanwhile, the optimal n_tree_ and m_try_ values for raw and SNV-SD datasets were re-selected, respectively. The selection process was the same as above. As shown in Appendix A, the 1527–1607 trees and 898–966 trees were the lowest ranges for n_tree_ of raw and SNV-SD leaf datasets, respectively. Then, 1570 and 900 trees were selected for the optimal n_tree_, as well as 18, and 18 were selected for the best m_try_ of the raw and SNV-SD datasets, respectively. Furthermore, these optimal n_tree_ and m_try_ were used to establish high-performance RF models. The error rate for the calibration set of raw datasets was reduced from 40.07% to 38.11%, and it decreased from 3.26% to 2.93% for the SNV-SD dataset. In addition, not only was the geographical origin classification ability of the RF model based on SNV-SD FT-MIR leaves spectra significantly better than that of the raw spectra, but higher performances were also obtained by the RF models of leaves than those of rhizomes. 

Parameters of sensitivity (SENS), specificity (SPEC), accuracy (ACC), and the Matthews correlation coefficient (MCC) for each class of calibration set and validation set of PLS-DA and RF model, based on raw and SNV-SD leaf FT-MIR spectra data matrices are displayed in Appendix A. Similar to the performance of parameters for the models of rhizomes, the values for all parameters of each class of calibration set and validation set for the PLS-DA model, based on raw leaf FT-MIR data matrices, were higher than that of the RF model. The identification ability of the SNV-SD PLS-DA model of the leaf data matrix almost reached the best ratings, and only samples collected from Yuxi and Dali were misclassified. Additionally, all parameters of validation set for the RF model based on the SNV-SD FT-MIR data matrixes were close, to the best, and only samples collected from Dali and Lijiang were misclassified. Additionally, parameters of two models for the SNV-SD data matrices based on FT-MIR spectra were higher than those of raw data matrixes. However, the classification performance for the PLS-DA and RF models on the basis of the leaf FT-MIR spectra were required for enhancement.

### 2.3. Regional Differences between VIP and Important Variables

The VIP and important variables of the RF and PLS-DA models of *P. yunnanensis* samples are displayed in Figure 7. In detail, Figure 7a,b are based on the raw FT-MIR spectra of rhizomes and leaves, respectively. Figure 7c,d are based on the SNV-SD FT-MIR spectra of rhizomes and leaves, respectively. The important variable numbers of the RF model of raw datasets for rhizomes and leaves were far more than those of the SNV-SD RF models. The variables with VIP values greater than 1 showed greater concentrations for several regions in the VIP scores based on raw rhizome and leaf matrixes, than those of the VIP scores of the SNV-SD datasets. From a comparison of the scatter of the most important variables between rhizomes and leaves, the number and distribution of important variables are different. It was demonstrated that various and different chemical profiles were contained between the rhizomes and leaves of *P. yunnanensis*. From the higher accuracy rate and the more uniform distribution of important variables of rhizomes or leaves in the PLS-DA model than those of rhizomes or leaves in the RF model, it was found that the PLS-DA was more suitable for the identification of geographical origins for *P. yunnanensis*. 

### 2.4. Data Fusion Strategy 

Despite the high performance obtained by PLS-DA, and the RF classification models of leaves of the FT-MIR spectra of *P. yunnanensis* samples, the 100% identification accuracy of the calibration set and the validation set were not acquired, and models’ abilities needed further enhancement. Hence, the data fusion strategy was used to further improve the prediction abilities of PLS-DA and RF models. Data fusion were concatenated variables of FT-MIR spectra from different parts, forming a single matrix where row numbers were the analyzed sample quantities, and columns consisted of variables. In other words, the rhizome and leaf datasets were combined to establish the classification models.

The process for establishing the data fusion RF model was similar to the individual dataset. RF models were established based on raw and SNV-SD data fusion FT-MIR spectra datasets. A total of 2414 and 2403 variables were contained in the two matrices, respectively. As shown in Appendix A, 356–399 trees and 375–404 trees were the lowest ranges for n_tree_ of raw and SNV-SD matrices, respectively. Additionally, the optimal values of 377 and 393 trees were selected for further selection of the suitable m_try_ values for raw and SNV-SD datasets, respectively. As shown in Appendix A, the optimal m_try_ were 51 and 18, respectively. Similar to the individual dataset, all variables were in ascending order with importance. The 10-fold cross-validation error rates of the RF model for raw and SNV-SD data fusion datasets are shown in Appendix A, respectively. Additionally, variable numbers of 69 and 288 with the lower 10-fold cross-validation error rates of 0.34853 and 0.02606 were selected to establish the data fusion RF models. Besides, the most important variables were the new data matrices, while re-selecting the optimal n_tree_ and m_try_ values for raw and SNV-SD data fusion datasets, respectively. As shown in Appendix A, the 1609–1660 trees and 98–125 trees were the lowest ranges for n_tree_ of the two datasets, respectively. Besides, 1652 and 104 trees, as well as 10 and 18, are selected for the best n_tree_ and m_try,_ respectively. Compared to the accuracy of the RF models between the raw and SNV-SD data fusion matrixes, the error rate for the calibration set of the raw dataset decreased from 37.46% to 37.13%, and decreased from 2.61% to 1.63% for the SNV-SD dataset. The classification abilities in the rhizome and lead data fusion RF model were better than in the individual dataset RF model. 

From a comparison of parameters for SENS, SPEC, ACC, and MCC between the PLS-DA and RF models, based on data fusion strategy, the PLS-DA model had a better classification ability than that of the RF model. As shown in Table 1, the geographical origins identification abilities reached the best of each class calibration set and validation set for the PLS-DA model of the SNV-SD FT-MIR spectra. However, the parameter values were close to 100% for most classes of RF model. Hence, it could be demonstrated that the PLS-DA model was more suitable for tracing the different geographical origins of cultivated *P. yunnanensis*. 

### 2.5. Hierarchical Clustering Analysis

HCA dendrograms based on average SNV-SD FT-MIR spectra datasets of rhizomes and leaves of *P. yunnanensis* from different geographical origins are presented in Figure 8a,b, respectively. It is obviously that all the six classes are grouping into two main clusters, both in the two HCA dendrograms. However, the clustering results among Kunming, Yuxi, Chuxiong, Dali, Lijiang and Honghe were obtained based on rhizomes and leaves FT-MIR spectral matrixes were different. As shown in Figure 9, the altitude is decreasing gradually from Northwest Yunnan to Southeast Yunnan. In addition, the two main clusters are influenced to some extent by the topography including altitude. Nevertheless, Kunming was cluster with Honghe and Yuxi in HCA dendrograms based on rhizomes dataset but cluster with Lijiang, Dali and Chuxiong of HCA plot based on leaves. It is demonstrated that the different chemical information between rhizomes and leaves of *P. yunnanensis* were influenced on the results of clustering.

## 3. Materials and Methods

### 3.1. Plant Material Preparation

In our experiment, rhizomes and leaves of 462 cultivated *P. yunnanensis* samples were collected from Kunming, Yuxi, Chuxiong, Dali, Lijiang, and Honghe cities in Yunnan Province; the collection locations and detailed information are shown in Figure 9 and Appendix A. All samples were identified as *P. polyphylla* Smith var. *yunnanensis* (Franch.) Hand.-Mazz. by Professor Hang Jin (Institute of Medicinal Plants, Yunnan Academy of Agricultural Sciences, China). To start with, the different parts for each *P. yunnanensis* samples were separated and washed, then dried at 50 degrees Celsius. In addition, both rhizome and leaf samples were sifted through 100 mesh sieves, and stored in a relatively dry environment.

### 3.2. FT-MIR Spectral Acquisition

FT-MIR analysis uses a FTIR spectrometer with a DTGS detector equipped, combined with a ZnSe attenuated total reflectance accessory (Perkin Elmer, Norwalk, CT, USA). The FT-MIR spectra collection parameters and methods are referenced in our previous experiment [14]. The FT-MIR spectra recorded ranges of 4000–550 cm^−1^ with 4 cm^−1^ resolution and 16 scans, both for rhizomes and leaves of each of the *P. yunnanensis* samples. Three scans were repeated for all rhizomes and leaves samples. Moreover, it was required that a relatively constant temperature and humidity was provided during the assessment of the FT-MIR spectra.

### 3.3. Chemometrics Methods

#### 3.3.1. Principal Component Analysis

PCA is an exploratory data analysis method and an unsupervised pattern recognition technique, which seeks for the optimum data distribution in a multivariate space [25,26,27]. The fundamental of PCA is that all the raw data are projected onto a two-dimensional sub-space, to ensure that information loss is minimized. The higher the front PCs, the higher the proportion of important variables represented. Generally, the first few PCs represent the most information. The first two or three PCs of all samples can be shown in two- or three-dimensional scores plots, and they further show the regularities of distribution for all the samples. Moreover, the relationship between the first two PCs and wave numbers can be shown by the loading plot.

#### 3.3.2. Partial Least Squares Discriminant Analysis

PLS-DA, a binary classification algorithm from 0 to 1, is based on the PLS algorithm, to add category labels to achieve the effect of classification prediction, and it shows the relationship by multivariate projection between independent and dependent variables, which are expressed by *X* and *Y*, respectively [28,29]. Besides, LVs were one feature variable that were produced by an intermediate process in the PLS-DA method [30]. LVs are useful for us to analyze the important variables and information. The *X* matrix and target and important values in *Y* are more closely correlated than the noise or unimportant values in *Y*. Additionally, the VIP plot summarizes the importance of the variables, both to explain *X*, and to correlate to *Y*, meaning that variables with a VIP value greater of than 1 are important; as well, those that are greater than 0.5 and less than 1 may be important, depending on the circumstances. Hence, classifying samples by PLS-DA requires that variables possess numbers that are greater than the classification sample numbers, and there should be some correlation among the identified samples.

#### 3.3.3. Random Forest

RF model, developed by Breiman in 2001, has been widely used to resolve classification problems in the field of food, and so on [31,32]. The RF model is based on the assembly classification or regression trees algorithm, and it shows a higher ability to resolve binary classification or regression issues [31]. The operational steps of the RF model can be roughly divided into the following five steps. Firstly, a spectra dataset was separated into two parts according to the ratio of 2 to 1, by the Kennard-stone (KS) algorithm by MATLAB 2017a (MathWorks, Natick, MA, USA) [33,34]. Two-thirds of the dataset was assigned as the calibration set (bootstrap samples), and one-third as the validation set (out-of-bag samples). The calibration set was used to obtain the optimal classification trees, and the validation set was applied to evaluate the ability of the FR model. Besides, the initial values of n_tree_ and m_try_ were defined as 2000, and the square root of the number of all variables, respectively. The optimal n_tree_ and m_try_ were both selected according to the lowest OOB classification error values. Thirdly, the most important variables were selected by a lower 10-fold cross-validation error rate, and as a new data matrix reimport. Fourth, the optimal n_tree_ and m_try_ were reselected according to the fundamental of step 2. Finally, the establishment of the final RF discrimination model was performed by using the optimized n_tree_ and m_try_ parameters_._ Step two to five were completed by R package (version 4.6–14).

#### 3.3.4. Hierarchical Cluster Analysis

HCA clusters different categories at a certain distance, according to the degree of similarity of each class, which means that it could preliminarily identify a classification trend for each category [35]. Besides, the Person correlation coefficient was applied to measure the linear relationship between the distance variables. These analyses were completed by SPSS 20.0 software (IBM Corp., Armonk, NY, USA).

### 3.4. Data Analysis

The purpose of data analysis involves the reduction of the influence by noise and other factors from experiments and instruments on the raw FT-MIR spectra data. Firstly, the raw FT-MIR spectra were pretreated by advancing ATR (attenuated total reflection) correction, and absorbance was transformed from transmittance by OMNIC 9.7.7 (Thermo Fisher Scientific, Madison, WI, USA). Secondly, the best preprocessing method was selected among a combination of various pretreatment methods, including SNV, FD and SD, which can enhance the accuracy and feasibility for identification study [36,37]. SNV and its derivatives could decrease a part of the irrelevant interferences, such as high frequency random noise, the interference of light scattering, baseline drift, and unequal concentration, and so on, to improve the classification ability of the models. All these preprocessing methods were completed by SIMCA-P^+^ 13.0 (Umetrics, Umea, Sweden). The datasets were separated into a calibration set and a validation set, with a rate of 2 to 1 by the KS algorithm, using MATLAB 2017a (The MathWorks), which was also used to establish the PLS-DA and RF models. In other words, the FTIR spectra of samples were divided into a calibration set (307 samples) and validation set (155 samples), as shown in Appendix A.

Generally, parameters including RMSEE, RMSECV, and the accuracy of calibration sets Q^2^ and R^2^ were used to estimate the identification ability of the calibration model [38,39]. The optimal preprocessing model required lower values of RMSEE and RMSECV, as well as higher values of accuracy for the calibration sets R^2^ and Q^2^. Besides, the model may have poor robustness and over-fitting when the values of the root mean square error of prediction (RMSEP) are greater than that of RMSECV [12]. In addition, due to both the PLS-DA and RF models being able to obtain the vote matrices, the two models could calculate the values of true negative (TN), true positive (TP), false negative (FN), and false positive (FP), respectively. SENS (Equation (1)), SPEC (Equation (2)), ACC (Equation (3)), and~MCC (Equation (4)) were the four parameters for each class, resulting in identification effects for different geographical origins of *P. yunnanensis* samples of PLS-DA and RF models. Obviously, this led to the higher values of these four parameters and a better identification ability for each class.
(1)$$\mathrm{SENS}=\frac{\mathrm{TP}}{\left(\mathrm{TP}+\mathrm{FN}\right)}$$
(2)$$\mathrm{SPEC}=\frac{\mathrm{TN}}{\left(\mathrm{TN}+\mathrm{FP}\right)}$$
(3)$$\mathrm{ACC}=\frac{\left(\mathrm{TN}+\mathrm{TP}\right)}{\left(\mathrm{TP}+\mathrm{TN}+\mathrm{FP}+\mathrm{FN}\right)}$$
(4)$$\mathrm{MCC}=\frac{\left(\mathrm{TP}\times \mathrm{TN}-\mathrm{FP}\times \mathrm{FN}\right)}{\sqrt{\left(\mathrm{TP}+\mathrm{FP}\right)\left(\mathrm{TP}+\mathrm{FN}\right)\left(\mathrm{TN}+\mathrm{FP}\right)\left(\mathrm{TN}+\mathrm{FN}\right)}}$$

## 4. Conclusions

In our article, the geographical origin identification of 462 *P. yunnanensis* samples from Kunming, Yuxi, Chuxiong, Lijiang, Dali, and Honghe were analyzed by rhizome and leaf FT-MIR spectra, combined with PLS-DA, RF, and HCA methods. The chemical information differences between rhizomes and leaves were directly displayed on the FT-MIR spectra and the results of models. PLS-DA was more suitable for use in classifying the different geographical origins of *P. yunnanensis* than the RF model, in that it had the best identification ability and more uniformly distributed important variables. Besides, the order of classification ability from strong to weak is the data fusion dataset > leaves dataset > rhizomes dataset, which means that leaves can be used quickly and accurately to identify the geographical origin of *P. yunnanensis*, and more comprehensive information can be showed by multiple sources of chemical information.

## Figures and Tables

**Figure 1 molecules-23-03343-f001:**
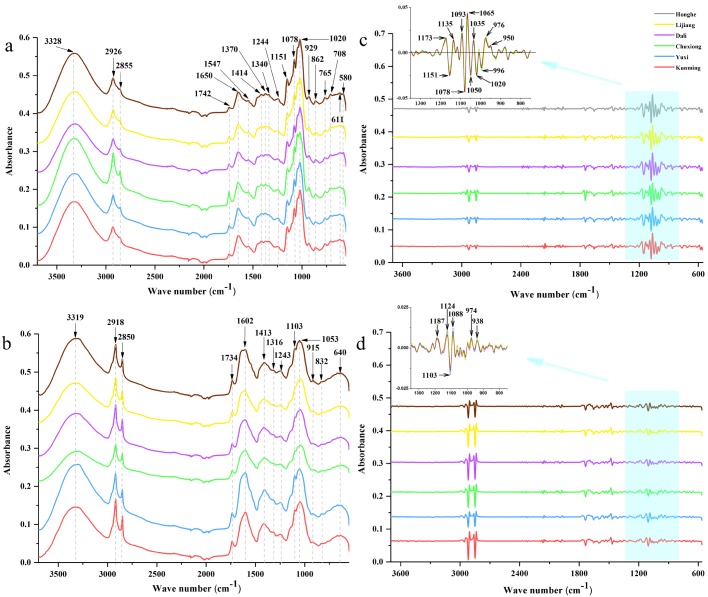
The FT-MIR spectra of Kunming, Yuxi, Chuxiong, Dali, Lijiang, and Honghe, Yunnan: (**a**) the raw spectra of rhizomes, (**b**) the raw spectra of leaves, (**c**) the best preprocessing spectra of rhizomes, (**d**) the best preprocessing spectra of leaves.

**Figure 2 molecules-23-03343-f002:**
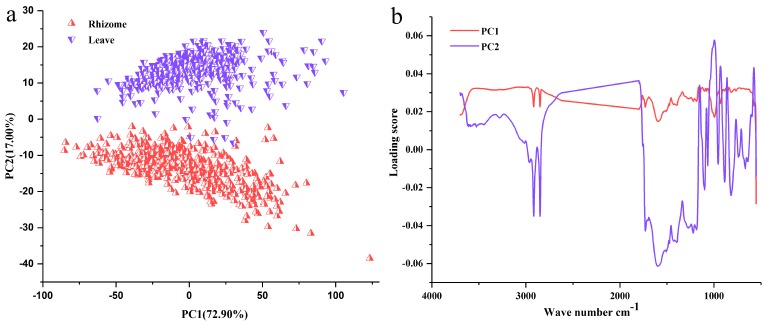
Principal component analysis (PCA) result based on Fourier transform mid infrared (FT-MIR) spectra: (**a**) Score plot, (**b**) Loading plot.

**Figure 3 molecules-23-03343-f003:**
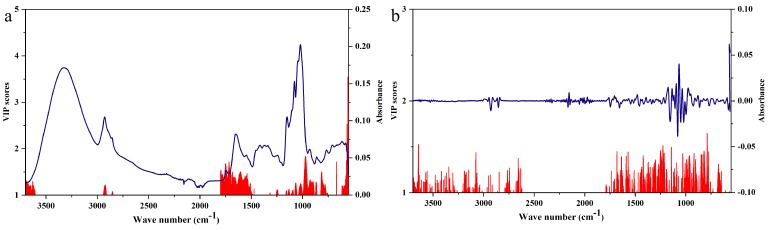
Variable importance for the projection (VIP) scores of the FT-MIR data of rhizomes for regional differences: (**a**) raw dataset, (**b**) standard normal variate–second derivative (SNV-SD) dataset.

**Figure 4 molecules-23-03343-f004:**
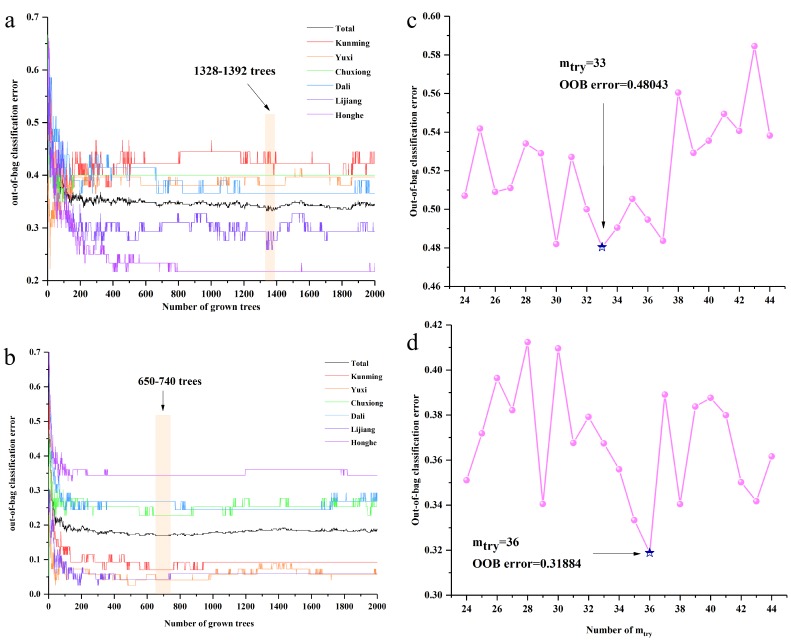
The n_tree_ and m_try_ screening of random forest (RF) models of *P. yunnanensis* samples before variables ranked by permutation accuracy importance: (**a**) n_tree_ of the raw rhizomes dataset, (**b**) n_tree_ of the SNV-SD rhizomes dataset, (**c**) m_try_ of the raw rhizomes dataset, (**d**) m_try_ of the SNV-SD rhizomes dataset.

**Figure 5 molecules-23-03343-f005:**
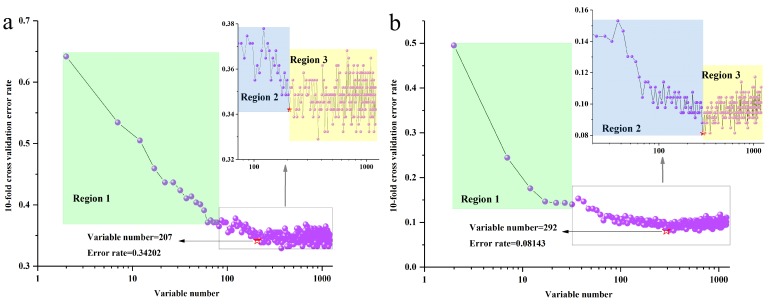
The 10-fold cross validation error rates of the RF model (sequentially reduce each five variables) based on *P. yunnanensis* samples: (**a**) raw rhizomes dataset, (**b**) SNV-SD rhizomes dataset.

**Figure 6 molecules-23-03343-f006:**
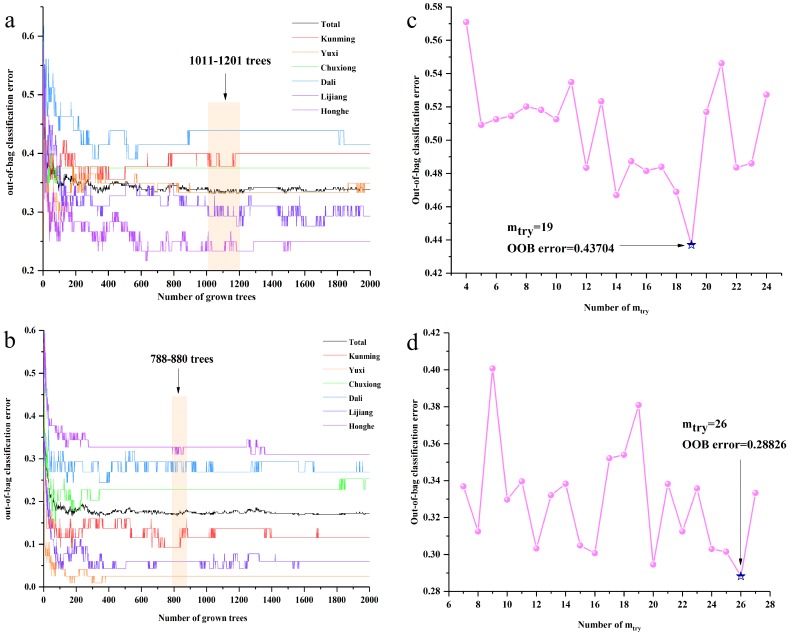
The n_tree_ and m_try_ screening of RF models of the *P. yunnanensis* samples after variables are ranked by permutation accuracy importance: (**a**) n_tree_ of the raw rhizomes dataset, (**b**) n_tree_ of the SNV-SD rhizomes dataset, (**c**) m_try_ of the raw rhizomes dataset, (**d**) m_try_ of the SNV-SD rhizomes dataset.

**Figure 7 molecules-23-03343-f007:**
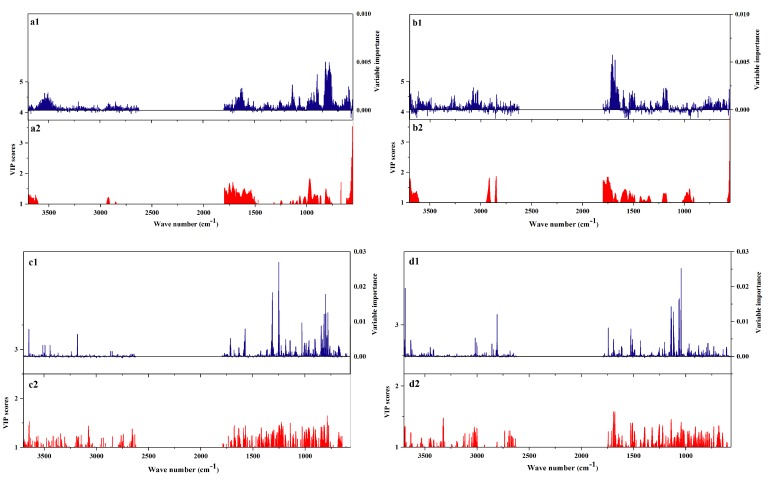
The importance variables (1) of RF models and the VIP values (2) of partial least squares discriminant analysis (PLS-DA) models of the *P. yunnanensis* samples: (**a**) the raw rhizomes dataset, (**b**) the raw leaves dataset, (**c**) the SNV-SD rhizomes dataset, (**d**) the SNV-SD leaves dataset.

**Figure 8 molecules-23-03343-f008:**
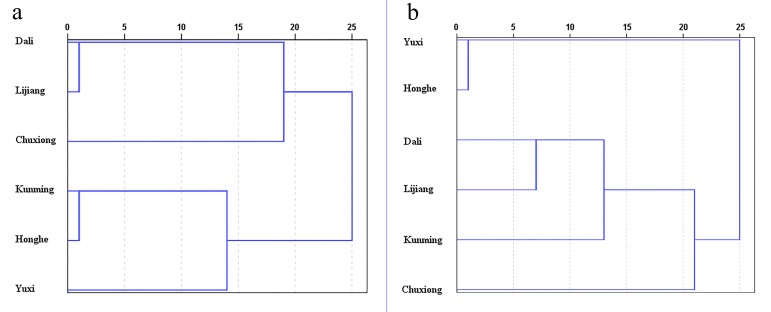
Dendrograms resulting of hierarchical cluster analysis (HCA) based on six geographical origins of *P. yunnanensis* samples: (**a**) the rhizomes dataset, (**b**) the leaves dataset.

**Figure 9 molecules-23-03343-f009:**
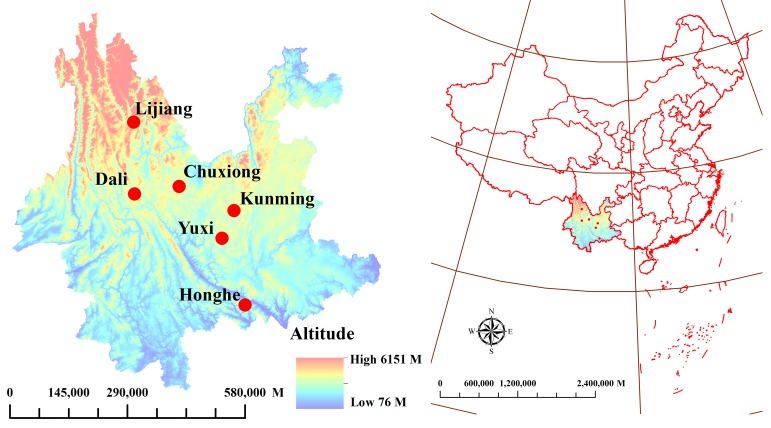
Location distribution of cultivated *P. yunnanensis* samples in Kunming, Yuxi, Chuxiong, Dali, Lijiang and Honghe, Yunnan Province.

**Table 1 molecules-23-03343-t001:** The major parameters of PLS-DA and RF models of each class, based on the data fusion SNV-SD FT-MIR spectra datasets of *P. yunnanensis* samples.

Preprocessing	Set	Classes ^a^	PLS-DA				RF			
			SENS	SPEC	ACC	MCC	SENS	SPEC	ACC	MCC
SNV-SD	Calibration set	1	1	1	1	1	1	0.996	0.997	0.987
		2	1	1	1	1	0.984	0.996	0.993	0.98
		3	1	1	1	1	0.975	1	0.997	0.986
		4	1	1	1	1	0.951	0.992	0.987	0.944
		5	1	1	1	1	0.9831	1	0.997	0.989
		6	1	1	1	1	1	0.996	0.997	0.99
	Validation set	1	1	1	1	1	1	1	1	1
		2	1	1	1	1	1	1	1	1
		3	1	1	1	1	1	1	1	1
		4	1	1	1	1	0.95	1	0.994	0.971
		5	1	1	1	1	1	0.992	0.994	0.979
		6	1	1	1	1	1	1	1	1

^a^ 1: Kunming, 2: Yuxi, 3: Chuxiong, 4: Dali, 5: Lijiang, 6: Honghe. Sensitivity (SENS), specificity (SPEC), accuracy (ACC) and the Matthews correlation coefficient (MCC).

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
