# Peer review of "Comparison and Identification for Rhizomes and Leaves of Paris yunnanensis Based on Fourier Transform Mid-Infrared Spectroscopy Combined with Chemometrics"

_molecules, 2018, doi:10.3390/molecules23123343_

Round 1

Reviewer 1 Report

I am sure that the Authors made a lot of effort to prepare this manuscript. However, I am not convinced that the quality of this work is high enough for the article to be published in the Molecules.

The major problem is that the Authors treat the spectra as a set of numbers and do not try to draw any conclusions based on their results in the view of chemical composition of the studied material. Further, there is no information provided about the exposition to the sun light of the places where the plants were cultivated (were there southern of norther slopes?).

Besides, the level of English is very low, there are many grammar and spelling mistakes, I present few of them below but the list should be much, much longer. The language quality is so poor that it makes the article unreadable.

line 18"is fluctuated" should be "fluctuates among"; "classification method was necessary to established" either "classification method was necessary to be established" or "it was necessary to establish..."

line 29: "described" not "prescribed"

lines 56-57: this sentence makes no sense

line 66 "quality" not "qualities"

The Authors should not start the sentence with "There is a saying...." It is a scientific article, not a story book!

line 75 "technologies"??? techniques! or methods!

Pages 24-32 are blank, is it a mistake or some data is missing?

Author Response

Dear Reviewer,

Thank you for your letter and reviewer’ comments concerning about our manuscript entitled “Comparison and identification for rhizomes and leaves of Paris yunnanensis based on Fourier transform mid infrared spectroscopy combined with chemometrics” (ID: molecules-407317). Those comments are valuable and very helpful for revising and improving our paper, as well as the important guiding significance to our researches. We have read comments carefully and have made corrections which we hope to meet with approval. Revised portion are marked in red font in the revised paper. The main corrections and the responds to the reviewer’s comments are displayed below:

Responds to reviewer1 comments:

Point 1: The major problem is that the Authors treat the spectra as a set of numbers and do not try to draw any conclusions based on their results in the view of chemical composition of the studied material. Further, there is no information provided about the exposition to the sun light of the places where the plants were cultivated (were there southern of norther slopes?).

Response 1: Thanks for your advice. Our main idea is that using differences of FT-MIR spectra among various samples to establish classification models for identify their geographical origin. Hence, we used the rhizomes and leaves FT-MIR spectra of P. yunnanensis to classify their geographical origin instead of by identifying their chemical composition. Besides, many successful cases of using spectral information to classification research of Chinese medicinal plants have been published, including Panax notoginseng [1], Curcuma longa [2], Radix Astragali [3], etc.

In addition, all cultivated P. yunnanensis samples were planted in sheds as shown in the Figure 1. The cultivated P. yunnanensis grows on the flat ground and the shed for cultivated samples was highlighted by a black circle.

1 Li, Y.; Zhang, J.Y.; Wang, Y.Z. FT-MIR and NIR spectral data fusion: a synergetic strategy for the geographical traceability of Panax notoginseng. Anal Bioanal Chem. 2018, 410, 91-103.

2 Gad, H.A.; Bouzabata, A. Application of chemometrics in quality control of Turmeric (Curcuma longa) based on Ultra-violet, Fourier transform-infrared and 1H NMR spectroscopy. Food Chemistry. 2017, 237, 857-864.

3 Hu, L.Q.; Yin, C.L.; Ma, S.; Liu, Z.M. Comparison and application of fluorescence EEMs and DRIFTS combined with chemometrics for tracing the geographical origin of Radix Astragali. Spectrochim Acta A. 2018, 205, 207-213.

Figure 1 The growing environment of cultivation P. yunnanensis samples

Point 2: Besides, the level of English is very low, there are many grammar and spelling mistakes, I present few of them below but the list should be much, much longer. The language quality is so poor that it makes the article unreadable.

line 18"is fluctuated" should be "fluctuates among"; "classification method was necessary to established" either "classification method was necessary to be established" or "it was necessary to establish..."

line 29: "described" not "prescribed"

lines 56-57: this sentence makes no sense

line 66 "quality" not "qualities"

The Authors should not start the sentence with "There is a saying...." It is a scientific article, not a story book!

line 75 "technologies"??? techniques! or methods!

Response 2: Thanks for your advice. The style, spelling and grammar of this manuscript were carefully checked out and revised into the right form.

Point 3: Pages 24-32 are blank, is it a mistake or some data is missing?

Response 3: Thanks for your advice. The contents of the 24-32 pages are figures 1 to 9, and these nine figures have been moved to the body part. Besides, these blank pages have been deleted.

Reviewer 2 Report

Thank you for the possibility to review this manuscript,, Comparison and identification for rhizomes and leaves of Paris yunnanensis based on Fourier  transform mid infrared spectroscopy combined with chemometrics. It covers a curious aspect with potential application character, especially for biostatics oriented reader, although in my opinion, this is rather theoretical work!. The authors should attempt a phytochemical evaluation of the studied plants?. The methodology is well and clearly described. Also results with discussion provide a concise and clear description of a core idea of the manuscript with confrontation of literature of the field (but it should be thoroughly analyzed). Authors also broadly presented possible practical applications of the study, what is undoubtedly a big strength of the manuscript.

Nevertheless, I have some minor comments to be addressed:

1.    The authors should be added from which family the plant comes from?

2.    Photos presented in manuscript are of a poor quality – please, attach these figures in better resolution.

3.    Keywords should be searchable by the exact word/phrase in the MESH library (https://meshb.nlm.nih.gov/search) it will improve visibility of the study in the internet and will be of benefit for future citations.

4.    The plant material was not described enough particularly -please add additional information.

Author Response

Dear Reviewer,

Thank you for your letter and reviewer’ comments concerning about our manuscript entitled “Comparison and identification for rhizomes and leaves of Paris yunnanensis based on Fourier transform mid infrared spectroscopy combined with chemometrics” (ID: molecules-407317). Those comments are valuable and very helpful for revising and improving our paper, as well as the important guiding significance to our researches. We have read comments carefully and have made corrections which we hope to meet with approval. Revised portion are marked in red font in the revised paper. The main corrections and the responds to the reviewer’s comments are displayed below:

Responds to reviewer 2 comments:

Point 1: The authors should be added from which family the plant comes from?

Response 1: Thanks for your advice. The plant comes from has been added as “The perennial herb plant Paris is a genus in the Liliaceae family.”

Point 2: Photos presented in manuscript are of a poor quality – please, attach these figures in better resolution.

Response 2: Thanks for your advice. Figures with better resolution have been anew supplied.

Point 3: Keywords should be searchable by the exact word/phrase in the MESH library (https://meshb.nlm.nih.gov/search) it will improve visibility of the study in the internet and will be of benefit for future citations.

Response 3: Thanks for your advice. Keywords have been revised as “Paris polyphylla Smith var. yunnanensis; multivariate analysis; chemometrics; Fourier transform infrared”.

Point 4: The plant material was not described enough particularly -please add additional information.

Response 4: Thanks for your advice. The detail information for the plant material has been added as shown in the Table1. And this table has been added in supplement materials as Table S5.

Table 1 The geographical location of P. yunnanensis samples

Region

Location

Sample size

Latitude (°N)

Longitude (°E)

1

Wuhua, Kunming

68

25.042165

102.704412

2

Hongta, Yuxi

95

24.43105

102.44098

3

Yaoan, Chuxiong

61

25.522293

101.375931

4

Weishan, Dali

61

25.307049

100.316085

5

Gucheng, Lijiang

87

26.874046

100.190409

6

Yuanyang, Honghe

90

23.007286

103.025416

Reviewer 3 Report

The work described in the manuscript could be suitable for publication on Molecules after major revisions.

The topic of the research is interesting and its relevancy in the literature is well described and contextualized. Nevertheless, in my opinion, few modifications are needed, in order to clarify the procedures followed to calculate the models, and to increase the general readability of the paper.

Major comments:

Comment 1: Despite the total number of analyzed samples is clear, I think it would be important for the reader to understand how the data set is organized. Consequently, I would suggest to inform the reader about the number of samples present into the training and the test sets, together with information about the class-belonging. I.e., I would like to know how many samples belong to each category; this information could be provided in a table or it could be mentioned in the text.

Comment 2: I really appreciate several details about the classification models are reported into the manuscript, nevertheless, in my opinion, the way they are presented is not completely correct. First of all, I would ask to the authors to clarify how the optimal preprocessing approach is chosen. In fact, they mentioned they looked at different parameters, but how they took into account all these parameters in order to compare the performances of the different models is not stated. For instance, I would suggest you to clarify why the SNV-SD has been considered the best pretreatment approach for PLS-DA on rhizomes. Did you take the average of the different parameters in order to compare the various models? Moreover, at page 6 lines 145 the authors say they used the RMSEP to choose the preprocessing approach. Model parameters and preprocessing approaches should be optimized on training samples and the test set should be left out until the final calibration model is built. Consequently, please clarify how you defined the optimal preprocessing approach but please be sure you did not use the test set to define it.  

Comment 3: As above-mentioned, the optimal pretreatment should be defined looking at cross-validated results, and then, once the optimal model is built, it is applied to the test set for the external validation. Consequently, I think it is not completely correct to show results on the test set independently from the pretreatment used. Consequently, I would suggest to modify all the tables in order to show all the cross-validated results (i.e., cross-validated results for all the models built on the pretreated data sets) and the results obtained on the test set only for the models built on data pretreated by the optimal preprocessing approach.
Additionally, I would also suggest the authors to add the number of latent variables used for the creation of the PLS-DA models.

Comment 4: Another major comment regards the data-fusion models. It is not completely clear to me how the spectra were organized prior the creation of the models, and how the data-fusion models were built. Please, provide more details about how the authors proceeded.   

 Comment 5: A further comment is about language; in my opinion, the general readability of the manuscript should be increased a bit. The entire text should be re-read by all the authors putting special attention on this regard. In particular, I would suggest the authors to re-write the sentences at the following lines:

·        Line 25.

·        From 68 to 70.

·        From 100 to 101.

·        From 122 to 124.

·        From 133 to 134.

·        From 192 to 193.

·        From 231 to 232.

·        From 294 to 297.

·        From 380 to 381.

Minor comments:

Several abbreviations (for instance: SNV-SD) are present into the manuscript. Despite many of them are explicated in Section 3.3, please, make sure the first time the reader meets an abbreviation, also the extended name is present.

Figure 1a: the peak at 3328 is indicated as 2926 in the figure. 

Page 15 Section 3.3.1: For PCA, please cite also:

1]K. Parson, On lines and plans of closes fit to systems of points in space, Philosophical

Magazine, 2 (1901) 559-572.

2] I.T. Jolliffe, Principal Component Analysis, second edition, Springer: New York, NY, 2002.

In section 3.3.1 the authors cited some papers where PCA was applied on similar problems as the one presented into the present manuscript. I appreciate this effort, because it helps contextualizing the work; nevertheless, I would suggest the authors to move these citations into the introduction, and use applications already present into the literature to motivate why you decided to use the methods you applied.   

Line 392 page 15 Section 3.3.2:  the reference 25 reported for PLS-DA is an application of this classifier; in this section it would be more appropriate to cite the original papers where PLS-DA was proposed. Consequently, I would suggest to mention also the following literature: 

1] M. Barker, W. Rayens, Partial least squares for discrimination, J. Chemometr. 2003, 17, 166_73.

2] L.Ståle, S.Wold, Partial least squares analysis with cross-validation for the two-class problem: a Monte Carlo study, J. Chemometr. 1987, 1, 185-196.

3]U.G.Indahl, H. Martens, T. Naes, From dummy regression to prior probabilities in PLS-DA, J. Chemometr. 2007 21, 529-536.

4] H. Nocairi, E.M. Qannari, E. Vigneau, D. Bertrand, Discrimination on latent components with respect to patterns. Application to multicollinear data, Comput. Stat. Data Anal. 2004, 48, 139-147.

Nevertheless, I appreciate you cited a work where the same technique was used with a similar aim. I would suggest you to include into the introduction few lines where you explain why you decided to use the chemometric approaches you used and where you could mention some works, already present in the literature, where the same methodologies have been used to classify natural products. In the literature, you will find several applications of PLS-DA or RF on MIR data, I would suggest you to look for some of them and to mention them together with [25]. Additionally, for a general overview over classification of plants you could also cite:

Biancolillo, A., Marini, F. (2018). Chapter Four - Chemometrics Applied to Plant Spectral Analysis. In J. Lopes, & C. Sousa (Eds.), Vibrational Spectroscopy for Plant Varieties and Cultivars Characterization, Comprehensive Analytical Chemistry, Volume 80 (pp. 69-104). Amsterdam: Elsevier.

Line 427 page 16 Section 3.3.3: Please, as already discussed above, move applications into the introduction and mention here the original paper where HCA was initially proposed.   

Table S1 and Table S3: What is the SMV mentioned in these tables? Is it SNV? In case, please correct it.

Author Response

Dear Reviewer,

Thank you for your letter and reviewer’ comments concerning about our manuscript entitled “Comparison and identification for rhizomes and leaves of Paris yunnanensis based on Fourier transform mid infrared spectroscopy combined with chemometrics” (ID: molecules-407317). Those comments are valuable and very helpful for revising and improving our paper, as well as the important guiding significance to our researches. We have read comments carefully and have made corrections which we hope to meet with approval. Revised portion are marked in red font in the revised paper. The main corrections and the responds to the reviewer’s comments are displayed below:

Responds to reviewer 3 comments:

Major comments:

Point 1: Despite the total number of analyzed samples is clear, I think it would be important for the reader to understand how the data set is organized. Consequently, I would suggest to inform the reader about the number of samples present into the training and the test sets, together with information about the class-belonging. I.e., I would like to know how many samples belong to each category; this information could be provided in a table or it could be mentioned in the text.

Response 1: Thanks for your advice. The sample size of calibration set and validation set for each class has been added as shown in the Table1. And this table has been added in supplement materials as Table S6.

Table 1 The sample size of calibration set and validation set for each class.

Class 1

Class 2

Class 3

Class 4

Class 5

Class 6

calibration set

45

63

40

41

58

60

validation set

23

32

21

20

29

30

Total

68

95

61

61

87

90

Point 2: I really appreciate several details about the classification models are reported into the manuscript, nevertheless, in my opinion, the way they are presented is not completely correct. First of all, I would ask to the authors to clarify how the optimal preprocessing approach is chosen. In fact, they mentioned they looked at different parameters, but how they took into account all these parameters in order to compare the performances of the different models is not stated. For instance, I would suggest you to clarify why the SNV-SD has been considered the best pretreatment approach for PLS-DA on rhizomes. Did you take the average of the different parameters in order to compare the various models? Moreover, at page 6 lines 145 the authors say they used the RMSEP to choose the preprocessing approach. Model parameters and preprocessing approaches should be optimized on training samples and the test set should be left out until the final calibration model is built. Consequently, please clarify how you defined the optimal preprocessing approach but please be sure you did not use the test set to define it.

Response 2: Thanks for your advice. The defined of optimal preprocessing approach has been revised in data analysis section as “Generally, parameters including root mean square error of estimation (RMSEE), root mean square error of cross-validation (RMSECV), accuracy of calibration sets, cumulative prediction ability (Q2) and cumulative interpretation ability (R2) were used to estimate identification ability of calibration model [1-2]. The optimal preprocessing model required the lower values of RMSEE and RMSECV as well as the higher values of accuracy of calibration sets, R2 and Q2.” Besides, the reason of SNV-SD was selected the optimal pretreatment method for rhizome datasets was revised in using rhizomes FT-MIR spectra datasets section as “Among them, SNV-SD was defined as the optimal preprocessing method to the fundamental for the larger values of R2, Q2, accuracy of calibration set as well as the lower values of RMSEE and RMSECV. Despite of SD obtained the better accuracy, SNV-SD obtained the lower RMSEE, RMSECV and LVs.” The reason of SNV-SD was selected the best preprocessing method for leaves datasets was revised in using leaves FT-MIR spectra datasets section as “Besides, the SNV-SD pretreatment among all preprocessing methods was the best one to classify the different origins of P. yunnanensis leaves samples, which possessed the satisfied values of R2, Q2, RMSEE, RMSECV, accuracy and LVs than other pretreatment methods.” The average of the different parameters was taken, and the example dataset of rhizome SNV-SD model as shown in Table2.

1        Xie, L.J.; Ye, X.Q.; Liu, D.H.; Ying, Y.B. Quantification of glucose, fructose and sucrose in bayberry juice by NIR and PLS. Food Chem. 2009, 114, 1135-1140

2        Qi, L.M.; Zhang, J.; Liu, H.G.; Li, T.; Wang, Y.Z. Fourier transform mid-infrared spectroscopy and chemometrics to identify and discriminate Boletus edulis and Boletus tomentipes mushrooms. Int J Food Prop. 2017, 20, S56-S68.

Table2 The average of the different parameters of rhizome SNV-SD model

RMSEE

RMSECV

RMSEP

Class 1

0.145713

0.199314

0.156293

Class 2

0.169756

0.22224

0.185843

Class 3

0.168353

0.221648

0.185739

Class 4

0.150649

0.211748

0.178837

Class 5

0.168519

0.2416

0.201246

Class 6

0.170237

0.221884

0.197242

Average

0.162205

0.219739

0.1842

Point 3: As above-mentioned, the optimal pretreatment should be defined looking at cross-validated results, and then, once the optimal model is built, it is applied to the test set for the external validation. Consequently, I think it is not completely correct to show results on the test set independently from the pretreatment used. Consequently, I would suggest to modify all the tables in order to show all the cross-validated results (i.e., cross-validated results for all the models built on the pretreated data sets) and the results obtained on the test set only for the models built on data pretreated by the optimal preprocessing approach.

Additionally, I would also suggest the authors to add the number of latent variables used for the creation of the PLS-DA models.

Response 3: Thanks for your advice. The table has been revised as shown in the Table3 and Table4. And these tables have been revised in supplement materials as Table S1 and Table S3.

Table 3 The major parameters of raw and preprocessing calibration models based on P. yunnanensis samples combined with rhizomes FT-MIR spectra.

Preprocessing

LVs

R2

Q2

RMSEE

RMSECV

Accuracy

Raw

18

0.714

0.584

0.201239

0.250368

95.44%

SNV

18

0.696

0.56

0.208316

0.254995

95.77%

SNV-FD

19

0.775

0.614

0.179515

0.239488

99.02%

SNV-SD

14

0.816

0.674

0.162205

0.219739

99.67%

SD

16

0.824

0.63

0.174812

0.237413

100%

Table 4 The major parameters of raw and preprocessing calibration models based on P. yunnanensis samples combined with leaves FT-MIR spectra.

Preprocessing

LVs

R2

Q2

RMSEE

RMSECV

Accuracy

Raw

18

0.698

0.559

0.203855

0.203855

95.77%

SNV

17

0.719

0.549

0.198488

0.256876

97.72%

SNV-FD

17

0.807

0.622

0.165141

0.240036

100%

SNV-SD

15

0.876

0.754

0.133634

0.195234

100%

SD

14

0.837

0.704

0.151242

0.210209

99.67%

Point 4: Another major comment regards the data-fusion models. It is not completely clear to me how the spectra were organized prior the creation of the models, and how the data-fusion models were built. Please, provide more details about how the authors proceeded.

Response 4: Thanks for your advice. The explanation for data fusion approach has been added in data fusion strategy as “Data fusion was concatenated variables of FT-MIR spectra from different parts, forming a single matrix that rows numbers was the analyzed samples quantities and columns were consisted by variables.”

Point 5: A further comment is about language; in my opinion, the general readability of the manuscript should be increased a bit. The entire text should be re-read by all the authors putting special attention on this regard. In particular, I would suggest the authors to re-write the sentences at the following lines:

Line 25.

From 68 to 70.

From 100 to 101.

From 122 to 124.

From 133 to 134.

From 192 to 193.

From 231 to 232.

From 294 to 297.

From 380 to 381.

Response 5: Thanks for your advice. The style, spelling and grammar of this manuscript were carefully checked out and revised into the right form.

Line 25: The obvious cluster tendency of rhizomes and leaves FT-MIR spectra was displayed by principal component analysis (PCA). The distribution of VIP was more uniform than important variables obtained by RF while PLS-DA models obtained higher classification abilities.

From 68 to 70: Complex climatic conditions in Yunnan, which means that quality of TCM plants varies with different climatic conditions of different geographical origins in Yunnan.

From 100 to 101: The peaks height, character and position among different geographical origins samples are similarly shown in Figure 1a.

From 122 to 124: Compared with raw rhizomes FT-MIR spectra, absorption for the raw leaves spectra exhibited red-shift at 1750-1290 cm-1 and blue-shift at 1290-950 cm-1. In other words, various difference of chemical information was reflected by the raw rhizomes and leaves FT-MIR spectra.

From 133 to 134: Two parts (rhizomes and leaves) were separated excellently by the first two PCs in PCA score plot. Absorption at 1300-550 cm-1 contributed to higher importance by PC 1 than PC 2. In other words, the bands of this region are more important to PC 2.

From 192 to 193: When the most important variables were re-selected forming the new data matrix, it was necessary for the reconstruction of optimal ntree and mtry values for raw and SNV-SD FT-MIR spectra. The selecting process was the same as above.

From 231 to 232: Besides, the SNV-SD pretreatment among all preprocessing methods was the best one to classify the different origins of P. yunnanensis leaves samples, which possessed the satisfied values of R2, Q2, RMSEE, RMSECV, accuracy and LVs than other pretreatment methods. For the following study of leaves, models established by raw and the best pretreatment (SNV-SD) FT-MIR spectra data were selected to study.

From 294 to 297: Comparison of the scatter of the most important variables between rhizomes and leaves, the number and distribution of important variables are different. It was demonstrated that various and difference chemical profiles were contained between rhizomes and leaves of P. yunnanensis.

From 380 to 381: The fundamental of PCA is that projected all the raw data on a two-dimensional sub-space and ensure the losing information to minimize.

Minor comments:

Point 1: Several abbreviations (for instance: SNV-SD) are present into the manuscript. Despite many of them are explicated in Section 3.3, please, make sure the first time the reader meets an abbreviation, also the extended name is present.

Response 1: Thanks for your advice. We have been added the extended name for abbreviation that appears for the first time.

Point 2: Figure 1a: the peak at 3328 is indicated as 2926 in the figure.

Response 2: Thanks for your advice. The Figure 1a has been anew supplied with the correct indication.

Fig 1 The ATR-FTMIR spectra of Kunming, Yuxi, Chuxiong, Dali, Lijiang and Honghe, Yunnan: (a) the raw spectra of rhizomes, (b) the raw spectra of leaves, (c) the best preprocessing spectra of rhizomes, (d) the best preprocessing spectra of leaves.

Point 3: Page 15 Section 3.3.1: For PCA, please cite also:

1]K. Parson, On lines and plans of closes fit to systems of points in space, Philosophical Magazine, 2 (1901) 559-572.

2] I.T. Jolliffe, Principal Component Analysis, second edition, Springer: New York, NY, 2002.

In section 3.3.1 the authors cited some papers where PCA was applied on similar problems as the one presented into the present manuscript. I appreciate this effort, because it helps contextualizing the work; nevertheless, I would suggest the authors to move these citations into the introduction, and use applications already present into the literature to motivate why you decided to use the methods you applied.

Response 3: Thanks for your advice. The applications of PCA have been moved to the introduction section and the references have been revised.

Point 4: Line 392 page 15 Section 3.3.2: the reference 25 reported for PLS-DA is an application of this classifier; in this section it would be more appropriate to cite the original papers where PLS-DA was proposed. Consequently, I would suggest to mention also the following literature:

1] M. Barker, W. Rayens, Partial least squares for discrimination, J. Chemometr. 2003, 17, 166_73.

2] L.Ståle, S.Wold, Partial least squares analysis with cross-validation for the two-class problem: a Monte Carlo study, J. Chemometr. 1987, 1, 185-196.

3]U.G.Indahl, H. Martens, T. Naes, From dummy regression to prior probabilities in PLS-DA, J. Chemometr. 2007 21, 529-536.

4] H. Nocairi, E.M. Qannari, E. Vigneau, D. Bertrand, Discrimination on latent components with respect to patterns. Application to multicollinear data, Comput. Stat. Data Anal. 2004, 48, 139-147.

Nevertheless, I appreciate you cited a work where the same technique was used with a similar aim. I would suggest you to include into the introduction few lines where you explain why you decided to use the chemometric approaches you used and where you could mention some works, already present in the literature, where the same methodologies have been used to classify natural products. In the literature, you will find several applications of PLS-DA or RF on MIR data, I would suggest you to look for some of them and to mention them together with [25]. Additionally, for a general overview over classification of plants you could also cite:

Biancolillo, A., Marini, F. (2018). Chapter Four - Chemometrics Applied to Plant Spectral Analysis. In J. Lopes, & C. Sousa (Eds.), Vibrational Spectroscopy for Plant Varieties and Cultivars Characterization, Comprehensive Analytical Chemistry, Volume 80 (pp. 69-104). Amsterdam: Elsevier.

Response 4: Thanks for your advice. The applications of PLS-DA and RF have been moved to the introduction section and the references have been revised. In addition, the explanation of chemometrics applications have been added in introduction section as “Up to now, Chemometrics has been widely applied to herbal medicines and plants spectral analysis [1-2]. For example, principal component analysis (PCA) often was used to research Chinese herbal medicines of multiple tissues and geographical origins [3-4]. Partial least squares discriminant analysis (PLS-DA) and random forest (RF) have been gradually applied to the field of traditional Chinese herbs in recent years such as Panax notoginseng, Dendrubium officinale, etc [5-6]

1 Gad, H.A.; El-Ahmady, S.H.; Abou-Shoer, M.I.; Al-Azizi, M.M. Application of chemometrics in authentication of herbal medicines: A review. Phytochemical analysis. 2012, DOI: 10.1002/pca.2378.

2 Biancolillo, A.; Marini, F. Chapter Four - Chemometrics Applied to Plant Spectral Analysis. In J. Lopes, & C. Sousa (Eds.), Vibrational Spectroscopy for Plant Varieties and Cultivars Characterization, Comprehensive Analytical Chemistry, Volume 80. Amsterdam: Elsevier. 2018, pp. 69-104.

3 Li, J.; Zhang, J.; Zhao, Y.L.; Huang, H.Y.; Wang, Y.Z. Comprehensive quality assessment based specific chemical profiles for geographic and tissue variation in Gentiana rigescens using HPLC and FTIR method combined with principal component analysis. Frontiers in Chemistry. 2017, 5, 125.

4 Qi, L.M.; Liu, H.G.; Li, J.Q.; Li, T.; Wang, Y.Z. Feature Fusion of ICP-AES, UV-Vis and FT-MIR for Origin Traceability of Boletus Edulis Mushrooms in Combination with Chemometrics. Sensors. 2018, 18(1). 241

5 Li, Y.; Zhang, J.Y.; Wang, Y.Z. FT-MIR and NIR spectral data fusion: a synergetic strategy for the geographical traceability of Panax notoginseng. Anal Bioanal Chem. 2018, 410, 91-103.

6 Wang, Y.; Huang, H.Y.; Zuo, Z.T.; Wang, Y.Z. Comprehensive quality assessment of Dendrubium officinale using ATR-FTIR spectroscopy combined with random forest and support vector machine regression. Spectrochim Acta A. 2018, 205: 637-648

Point 5: Line 427 page 16 Section 3.3.3: Please, as already discussed above, move applications into the introduction and mention here the original paper where HCA was initially proposed.

Response 5: Thanks for your advice. The applications of HCA have been moved to the introduction section and the reference 35 in manuscript have been revised.

1 Jain, A.K.; Dubes, R.C. Algorithms for clustering data. Technometrics, Prentice-Hall, Inc. Englewood Cliffs, New Jersey. 1988.

Point 6: Table S1 and Table S3: What is the SMV mentioned in these tables? Is it SNV? In case, please correct it.

Response 6: Thanks for your advice. The SMV has been revised as SNV. Besides, the exact language of this manuscript and supplement materials was carefully checked out and revised.

Round 2

Reviewer 1 Report

The Authors have answered on most of my questions, still in my opinion the level of English should be improved.

Reviewer 3 Report

I would like to thank the authors for the effort they put in the revision. 
I am glad to suggest to accept you paper for publication.